# Krukenberg Progression of Gastric Carcinoma in Pregnancy: Is Early Diagnosis Possible? Case Report and Review of the Literature

**DOI:** 10.3390/jcm12165397

**Published:** 2023-08-19

**Authors:** Fanny Eckel, Greta Carlin, Stefanie Mayer, Stephan Polterauer, Kinga Chalubinski

**Affiliations:** 1Department of Obstetrics and Gynecology, Division of Obstetrics and Feto-Maternal Medicine, Medical University of Vienna, 1090 Vienna, Austria; 2Department of Obstetrics and Gynecology, Division of General Gynecology and Gynecologic Oncology, Medical University of Vienna, 1090 Vienna, Austria

**Keywords:** Krukenberg, ovarian cancer, ovarian tumor, ovarian metastasis, pregnancy, gastric cancer

## Abstract

Krukenberg tumors are metastatic tumors of the ovaries, associated with poor outcomes. Most commonly, these tumors are of gastric origin. The diagnosis of Krukenberg tumors in pregnant patients is extremely rare and poses specific difficulties for clinicians. We report a case of a pregnant woman presenting with an unknown abdominal tumor. Through the use of magnetic resonance imaging, multiple differential diagnoses were proposed, including a malignant ovarian tumor. A cesarean section and explorative laparotomy were conducted, revealing Krukenberg metastases of a gastric tumor, discovered during intraoperative gastroscopy. Tumor resection with concomitant chemotherapy was conducted. The main aim of this paper was to evaluate whether earlier diagnosis seems possible in such cases. A thorough literature review was conducted, unfortunately revealing no reliable method for early detection. Furthermore, no consensus regarding diagnostics or therapy exists to date. Thus, more research should be conducted regarding this rare condition to offer recommendations regarding early detection, diagnostics, and therapeutic approaches.

## 1. Introduction

Adnexal masses can be found in 2.3 to 4.1% of all pregnancies during the first trimester. Approximately 90% of these cases resolve by themselves and do not cause any problems for the woman or fetus [1]. Most adnexal masses in pregnancy are benign; the rare diagnosis of ovarian cancer complicates approximately four to eight out of 100,000 pregnancies [2]. The most common type are epithelial tumors [2]. Krukenberg tumors in pregnancy are rare and are associated with a poor outcome [3].

A Krukenberg tumor is a rare type of glandular carcinoma which metastasizes to the ovaries. It accounts for approximately 1–2% of all ovarian tumors and up to 17.8% of all ovarian cancers [4]. Different primary cancer sites have been described, such as the stomach, appendix, colon, breast, small intestine, rectum, gallbladder, and urinary bladder. The most common primary tumor leading to Krukenberg metastasis is gastric cancer, accounting for two thirds of all cases [5]. A few case reports describe Krukenberg tumors in patients where no primary tumor could be found. Nevertheless, it remains unclear whether primary Krukenberg tumors exist [4,6].

The aim of this paper is to present the difficulties in the diagnosis and treatment of pregnant patients diagnosed with Krukenberg tumors based on a case report of a female patient recently diagnosed and treated at our institution. Furthermore, we aim to offer an outlook on possibilities for the early detection of these cases according to a thorough review of the literature.

## 2. Case Presentation

A 38-year-old Caucasian woman was antenatally referred to our institution by a primary care hospital with a timely developed singleton pregnancy in 35 + 1 under the suspected diagnosis of placenta percreta. It was the patient’s second pregnancy, after a cesarean delivery 2 years prior. She had no history of endometriosis, previous abdominal operations, or other relevant medical conditions. Regarding the patient’s family history, her mother suffered from a type-one endometrial carcinoma, while an aunt was diagnosed with breast cancer at the age of 55.

Clinical examination upon admission showed a completely asymptomatic patient. Ultrasound imaging showed a timely developed singleton pregnancy, and an invasive placental growth seemed unlikely as the intrauterine placentation showed no sonomorphological placenta accreta spectrum abnormalities. However, in the right middle abdomen, adjacent to the uterine wall, a well-definable, echo-dense tumor of 99 × 73 × 75 mm in size could be detected, with surrounding ascites, which is shown in Figure 1. A comparable structure was also displayed on the left side, measuring 70 × 34 × 37 mm, most likely originating from the left adnexa. Both lesions showed an area with increased vascularization. Sonographically, offering a definite diagnosis was not possible, but the initial diagnosis of placenta percreta was reassigned as adnexal tumors, possibly chorion tumors.

Thus, magnetic resonance imaging (MRI) was performed for further diagnostic clarification. Fetal MRI confirmed the pre-determined finding of the ultrasound, and the abdominal MRI showed an inhomogeneous formation, measuring 70 × 130 × 63 mm and located between the liver and uterus, as depicted in Figure 2. In addition, several individual, questionably communicating structures in a total extent of about 58 × 29 × 71 mm diameter of analogous morphology could be detected in the left lower abdomen, some of which could not be well distinguished from the adjacent intestine. The structures primarily emanated from the adnexa. Furthermore, moderate ascites, especially in the pelvis minor with questionable, small, nodular changes in the cavum Douglasi were visible. At this point, there were three differential diagnoses: granulosa cell tumors, decidualized endometriomas, and malignant neoplasia.

Laboratory testing showed no abnormalities in the blood count; however, serum levels of protein and albumin were low and liver function parameters were elevated. Furthermore, tumor marker CA-125 (cancer antigen) was increased to 324 kU/L, while CA-19-9, CA 15-3, and CEA (carcinoembryonic antigen) were in the norm. AFP (alpha-fetoprotein) and beta-2-microglobulin levels were also elevated (AFP: 37.7 IU/mL; beta 2-microglobulin: 2.61 mg/L).

The following treatment plan was decided on in accordance with the patient’s wishes: continuation of the pregnancy up to 38 weeks, followed by delivery via cesarean section with additional right adnexectomy for intraoperative frozen section diagnosis. Depending on its findings, the further extent of the operation would be decided.

However, due to a sudden aggravation of the patient’s symptoms, specifically reduced general condition, nausea, emesis, and abdominal pain, the surgery took place sooner than originally planned, at 36 + 3 gestational weeks. After an uneventful cesarean section, a thorough inspection revealed a large right adnexal tumor. Thus, the right adnexa were fully resected; the surgical specimen is shown in Figure 3. Frozen section diagnosis showed metastases of mucinous adenocarcinoma in the ovary and tube wall, morphologically compatible with Krukenberg metastasis of a gastrointestinal tumor.

Therefore, additional left adnexectomy was performed. The left adnexa presented with macroscopically visible metastases and was measured at 75 × 45 × 30 mm. Afterwards, another thorough inspection of the pelvic cavity occurred. A tumor infiltration in the bladder wall was visible; thus, partial bladder resection and reconstruction were performed. Further exploration of the abdomen showed carcinosis on the appendix, the anterior rectum wall, the transverse colon, the small intestine mesentery, and multiple sites of the small intestine. Due to small-nodular peritoneal carcinosis, a total peritoneal carcinosis index of about 15 was found. Small-nodular carcinosis could be seen on the right diaphragm and spleen. Furthermore, a tumor, approximately 5 cm in size, in the distal stomach was visualized. This was suspected to be the primum; therefore, an intraoperative gastroscopy was conducted. The gastroscopy showed a large-scale infiltration by foreign tissue in the anterior stomach wall, and several biopsies were taken.

The histological examination further showed metastasis of mucinous adenocarcinoma, morphologically well compatible with the known gastric primum in both fallopian tubes, the left ovary, omental and intestinal biopsies, and bladder peritoneum. In the conducted immunohistochemical diagnosis, no evidence of microsatellite instability was found, the PD-L1 tumor proportion score/TPS was 0% and the combined positive score/CPS was 1%, the pan-TRK was negative, and CISH with probes against HER-2 and Centromer 17 (Ventana, Oro Valley, AZ, USA) showed no amplification of the HER-2 gene. DPYD diagnostics showed no evidence of the presence of DPYD*2, *13, rs67376798, or HapB3.W normal metabolizer. Unfortunately, no significant target markers could be identified.

Due to the diagnosis of a stage-four adenocarcinoma, staging computer tomography (CT) was performed, which showed an elongated, diffusely enhanced, and thickened stomach wall, matching a diffuse gastric carcinoma, suspicious peri-gastric lymph nodes, moderate ascites in all four abdominal quadrants with reactive contrast-enhanced peritoneum, and extensive pleural effusions on both sides.

The administration of systemic chemotherapy according to the FLOT scheme (5-fluorouracil, leucovorin, oxaliplatin, and docetaxel) was planned, with a reduction to FOLFOX (combination chemotherapy regimen including 5-fluorouracil, leucovorin, and oxaliplatin) in the course, depending on the patient’s general condition and tolerability.

The first restaging CT scan after six cycles of chemotherapy showed partial remission. Currently, the patient continues to receive palliative chemotherapy.

## 3. Discussion

Krukenberg tumors are an extremely rare complication in pregnancy, associated with a poor outcome [3]. A review conducted in 2016 showed a median survival time of six months in this patient collective [7]. Up to now, there is no consensus on the necessary further diagnostic measures and treatment options in pregnant patients with Krukenberg tumors [7].

In our case, a malignant tumor deriving from the left adnexa was suspected after ultrasound and MRI scanning. Only after surgical resection, histological examination, and intraoperative gastroscopy could the diagnosis of a Krukenberg metastasis of a gastric carcinoma be confirmed. There are multiple case reports of Krukenberg tumors diagnosed in pregnant women [8,9,10]. To describe and compare all reported cases would be too extensive for this paper; therefore, the most relevant articles were chosen. However, initial misdiagnosis based on imaging techniques appears to be a common problem often described in the literature. Reported differential diagnoses include epithelial ovarian cancer [3], benign tumors [11], Meigs’ syndrome [12], or luteoma [13]. The main concern is that this often leads to a delay in treatment since surgical resection of a suspected benign tumor in pregnancy is often postponed to avoid unnecessary pre-term birth. Many of the described cases were only operated on soon after the initial tumor diagnosis due to acute situations such as ovarian torsion or pathological cardiotocography [11,12,13]. In our case, surgery was performed six days after the initial suspected diagnosis of ovarian cancer. Intraoperative gastroscopy and postoperative staging computer tomography were conducted. In the literature, widely different approaches regarding further diagnostic measures have been described. There are cases where no further evaluation was conducted due to missing consequences in the palliative setting as well as due to the patient’s wishes [11], while the entire possible diagnostic work-up, including full endoscopy and multiple imaging techniques such as positron emission tomography, was utilized in other cases [3].

Regarding management after definite diagnosis, once more, no consensus can be found in the literature. With poor outcomes and a median survival rate of six months [7], different approaches have been described. Magudapathi et al. describe a case where a patient underwent staging laparotomy as well as three cycles of palliative chemotherapy during pregnancy before going into pre-term labor. After delivery, the patient received further chemotherapy but nevertheless died four months later [14]. In another case, antepartum chemotherapy was discussed as a possible therapy option but was finally not administered owing to risks for the fetus as well as the mother, especially due to elevated liver function parameters in the course of preeclampsia [15]. In a case described by Mendoza et al., a young woman diagnosed with a Krukenberg tumor during pregnancy underwent a cesarean section and an exploratory laparotomy. Due to the excessive non-resectable tumor extent, the patient did not receive chemotherapy but rather only symptomatic palliative treatment and died two months later [16]. In another case report, published by Zhang et al., a C-section and bilateral adnexectomy were performed at 38 weeks’ gestation. As soon as the primary tumor was found to be of gastric origin, the patient underwent total gastrectomy and intraperitoneal chemotherapy. However, no further follow-up regarding this patient was reported [3]. These are only a few examples of possible approaches to treat women with this rare and fatal diagnosis. In our case, we conducted a cesarean section combined with cytoreductive surgery and subsequent palliative chemotherapy, which the patient receives to date, around three months after the initial diagnosis.

It can be summarized that, regarding different aspects including clinical presentation, diagnostics, and therapeutic approaches, the individual cases shown through a variety of case reports have widely heterogenous developments and are therefore difficult to summarize [8,9,10].

The tragic nature of a fatal diagnosis in pregnant women, endangering the life of not only the mother-to-be but also her fetus, begs the question of whether an earlier diagnosis could be achieved. Thus, this question was defined as the main aim of this paper, and a thorough literature search was conducted to find suggested methods of early detection. Ideally, the primary tumor would have to be diagnosed and treated before metastases occur. Most Krukenberg tumors derive from gastrointestinal primary tumors, most commonly gastric carcinoma [5]. The typical symptoms caused by gastric carcinoma are vomiting, abdominal fullness, epigastric pain, and weight loss. However, all of these conditions can result from physiological changes during pregnancy and thus do not usually worry patients or clinicians [13]. Scharl et. al. reported that if all pregnant women with relevant nausea and emesis underwent gastroscopy, about 1,000,000 gastroscopies would be needed to diagnose approximately 9 cases of gastric carcinoma [17]. Another factor that makes early diagnosis in these cases difficult is the teratogenic effect of radiation, which almost completely prohibits standard imaging techniques like computer tomography of the abdominal area during pregnancy. Another commonly used diagnostic tool that is not as easily applicable in pregnancy are tumor markers. While CEA, CA 19-9, and CA 15-3 are not affected, other markers such as AFP or CA-125 are physiologically elevated during pregnancy [18]. Additionally, the progression of gastric carcinoma can be accelerated during pregnancy due to multiple reasons. The physiologically increased sex hormone levels in pregnant women can stimulate the further development of gastric precancerous lesions, as these are known to be able to express estrogen receptors [13]. Furthermore, high levels of placental growth factors can be detected in gastric cancer tissue; this factor is associated with negative prognostic parameters such as serosal invasion or lymph node metastasis [16].

Therefore, taking into account the described difficulties, we had to conclude that there is no reliable option for early diagnosis in these cases. The only suggestion that can be made regarding this problem is to interpret gastrointestinal symptoms such as nausea, emesis, or melena as warning symptoms if they occur in a way that is not typical for the regular side effects of pregnancy. While nausea and vomiting are typical in the first trimester, they should be further evaluated when first appearing in the second or third trimester—melena during pregnancy, on the other hand, should generally lead to further diagnostic evaluation [3].

## 4. Conclusions

Krukenberg metastasis in pregnancy is a rare diagnosis with a median survival of six months [7]. The most common primary tumor is gastric carcinoma. After a thorough evaluation of the existing literature, we came to the conclusion that, to date, there is no reliable and feasible method for the early screening and diagnosis of gastric cancer in pregnancy. However, a history of gastric ulcers or bleeding, as well as new occurrences of nausea and vomiting after the first trimester of pregnancy should be considered as alarming symptoms and thus lead to further evaluation [3]. Further research needs to be conducted in this field to offer standardized procedures regarding diagnostics and therapy.

## Figures and Tables

**Figure 1 jcm-12-05397-f001:**
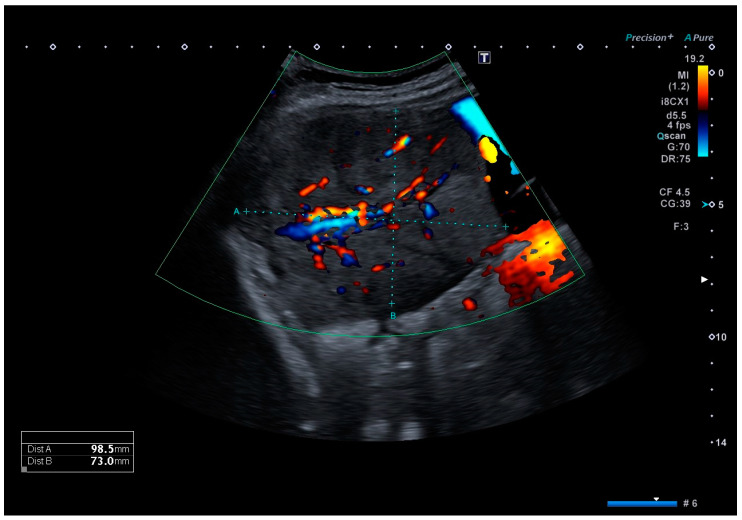
Color Doppler ultrasound image of the tumor located in the right middle abdomen, depicting a moderate vascular flow. The tumor is embedded in ascites.

**Figure 2 jcm-12-05397-f002:**
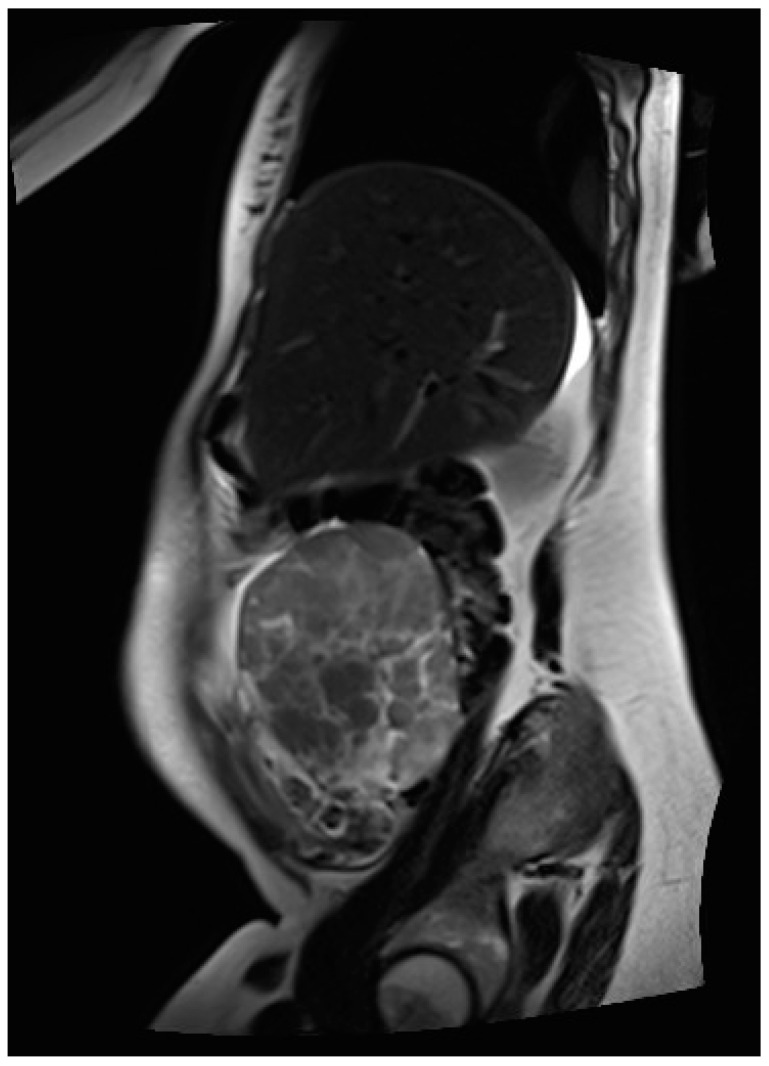
MR image of the inhomogeneous, mainly solid tumor located in the right middle abdomen, primarily deriving from the right adnexa.

**Figure 3 jcm-12-05397-f003:**
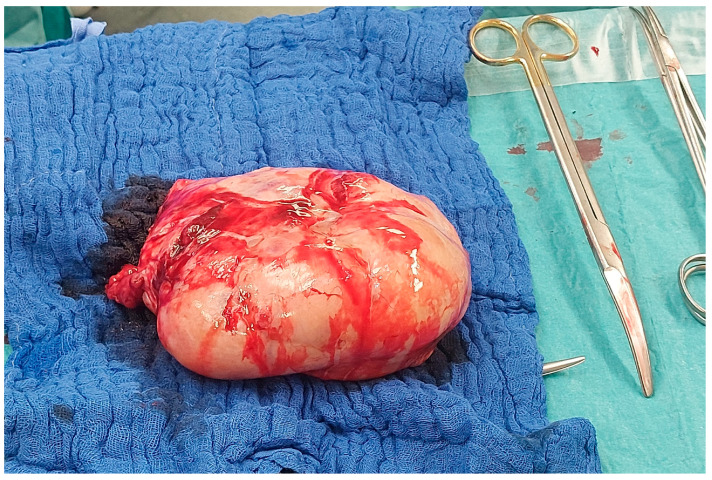
Photograph of the surgical specimen of the excised right adnexa during the operation.

## Data Availability

The data presented in this study are available from the corresponding author upon request. The data are not publicly available due to privacy considerations.

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
