# Peer review of "Krukenberg Progression of Gastric Carcinoma in Pregnancy: Is Early Diagnosis Possible? Case Report and Review of the Literature"

_jcm, 2023, doi:10.3390/jcm12165397_

Round 1

Reviewer 1 Report

First, I would like to congratulate the authors for their work. I've seen a few cases of Krukenberg in association with pregnancies and these are indeed very difficult cases from every single perspective. Therefore, I strongly believe that this subject may be of interest for the readers of this journal.

 However, I also believe that the manuscript requires some revisions:

1.      Row 27- according to the rules of formal writing, one should not start a sentence with a number. Please try to reformulate.

2.      Please add more citations, for “Successful Feto-maternal Outcome Following Pregnancy in a Patient with Krukenberg Tumor: Case Report and Review of Literature”, as well as “Primary Krukenberg tumour in pregnancy”

3.      Can you please also elaborate on the pathologic aspect of the tumor? Was it a signet ring cell carcinoma? Was the tumor cystically transformed? Was immunohistochemistry performed? What size did the left ovarian metastasis have? Was it also macroscopically identifiable?

English writing is good but requires proofreading for minor grammatical errors before publishing.

Reviewer 2 Report

This manuscript reported the case of Krukenberg tumors metastasized from gastric cancer in pregnant patients and its review. The authors concluded that there was no reliable and feasible method for early screening and diagnosis of gastric cancer. Therefore, gastrointestinal symptoms will be alarming symptoms and it will lead to further evaluation.

This manuscript is well organized; however, I suggest some revisions for a better understanding of this manuscript as follows.

               Line 90, What was a sudden aggravation of symptoms which lead to the surgery?

               Were there any symptoms such as edema or an excessive increase in abdominal circumference which might be caused by the pooling of ascites and the growth of tumors?

Reviewer 3 Report

The authors presented an interesting case of Krukenberg tumor diagnosed during pregnancy. however, there is no new information regarding early detection or treatment. 

Quality of English language is excellent. 

Author Response

Thank you for your consideration. There are no comments to adress. 

Round 2

Reviewer 3 Report

The new version added molecular testing for the gastric tumor is appreciated. It is disappointing that no significant target markers were found, which is common in these tumors. Detailed case reviews were provided for comparison.

Minor editing especially medical terminology.

Author Response

Thank you for your comment. A short statement regarding the absent target markers in this case has been added to the case presentation. We did not go into further detail, since the focus of this paper is supposed to be early diagnosis rather than treatment details. As the reviewer mentioned, multiple case reports are already mentioned for comparison.